# Association between mental illness and disciplinary confinement and its effect on mental health: A systematic review and meta-analysis

Sabrina Giguère [1,2], Laura Dellazizzo [1,3], Charles-Édouard Giguère [2], Alexandre Dumais [1,2,4]*

1 Department of Psychiatry and Addictology, University of Montreal, Montreal, Canada, 2 Research center of the University Institute in Mental Health of Montreal, Montreal, Canada, 3 Medicine, University of Montreal, Montreal, Canada, 4 Institut national de psychiatrie légale Philippe-Pinel, Montreal, Canada

* alexandre.dumais@umontreal.ca

## Abstract

The use of disciplinary confinement (DC) as a form of punishment in detention raises international attention and many concerns that have led to the elaboration of several studies. However, as studies report mixed results regarding the risk of DC placement for mentally ill inmates and the possibility of psychological effects from exposure to DC, it was necessary to shed some light in order to better guide future decisions, policies and programs in detention. Thus, this systematic review and meta-analysis aimed to evaluate the risk of inmates with mental disorders being placed into disciplinary confinement and its effect on mental health. A systematic search of studies was performed in PubMed, PsycINFO, Web of Science, and Google Scholar. The meta-analysis was conducted using random-effects models. Heterogeneity among study point estimates was assessed with Q statistics and quantified with the $I^2$ index. Publication bias was assessed using Egger's test. Quality assessment was based on the GRADE Checklist for observational studies. Guidelines from Preferred Reporting Items for Systematic Reviews and Meta-Analyses (PRISMA) were followed. First, a meta-analysis of five articles including 27,455 inmates showed that incarcerated individuals with a mental disorder were 1.23 times (OR=1.23, CI = 1.10; 1.38) more likely to be placed in DC than incarcerated individuals without a mental disorder. Notably, having a severe mental disorder (OR=1.31, p < 0.001), a personality disorder (OR=1.66, p < 0.001), and having previously received mental health services (OR=1.16, p = 0.024) increased the risk of being placed in DC. Secondly, a systematic review of six articles, including 171,300 inmates, showed more psychological distress, psychiatric symptoms (self-harm, thought disorders, obsessive-compulsive symptoms), need for mental health services, and hospitalizations in DC than in the general correctional population. Considering the increased risk of placement in DC for incarcerated persons with a mental disorder and its deleterious effect on mental

**Data availability statement:** All relevant data are within the manuscript and its Supporting Information files.

**Funding:** The author(s) received no specific funding for this work.

**Competing interests:** The authors have declared that no competing interests exist.

state, it is essential that new safe interventions to manage these inmates are created to limit the use of DC and offer them proper mental health care. These results align with the ongoing concerns regarding the use of DC in correctional settings, which has raised widespread international attention on public health and human rights, and they provide critical insights for policy reforms and better practices in correctional facilities worldwide.

## Introduction

People with mental disorders are overrepresented in correctional facilities [1–3]. Indeed, 47.7% of individuals in an American prison in 2015 had a mental disorder; this number went up to 74.6% in a study that included 10 Canadian jails in 2017 [1,2]. Moreover, a national survey in 2004–2005 showed that in some areas of the United States of America (e.g., Arizona, Nevada), a higher prevalence of individuals with a mental disorder was in correctional facilities than in psychiatric hospitals [4]. However, compared to psychiatric hospitals, correctional facilities have fewer appropriate resources and services for mental disorders, and training for correctional staff is very limited [2,5]. Thus, psychiatric symptoms may be mistaken for an intentional act of disobedience by staff [6,7]. In addition, individuals with mental disorders have more difficulty complying with regulations (e.g., strict schedules), and they adapt less well to the correctional environments (e.g., by committing auto and hetero aggressive behavior) [3,6,7]. In correctional facilities whose primary objective is to preserve institutional order and security, one of the ways to manage, prevent, and control disruptive behavior is through the use of solitary confinement [7–10].

Solitary confinement exists in three forms with different objectives: 1) administrative confinement for institutional management purposes (e.g., inmates who fail to appropriately adjust in the general carceral population by violating facility rules), 2) protective confinement for the inmate's personal safety, and 3) disciplinary confinement (DC) as punishment for failure to comply with an institutional rule [9–11]. Concerning the latter, which is within the scope of this paper, a disciplinary process governs this type of solitary. When an inmate is ticketed by a guard for a rule violation, they must appear before a disciplinary committee to determine the punishment associated with the violation. Various punishments are available, from a warning to the most severe disciplinary measure being DC [12]. Generally, inmates in solitary confinement are kept in their cells for 23 hours a day, with one hour for physical exercise and hygiene care. These cells are generally physically separated from the general correctional population. Access to programs and services (e.g., education, rehabilitation), recreation, and human socialization are very limited [13–16]. In some cases, the cells may be lit continuously with artificial light that inmates cannot control, and there may be no access to natural light [17].

This isolated and disenfranchised environment has given rise to several studies into the repercussions on inmates' mental state, leading to a few meta-analyses. A first quantitative synthesis on administrative confinement specifically found a

significant small to moderate effect on anxiety (g = 0.39, CI = 0.08; 0.70) and general health (g = 0.61, CI = 0.14; 1.08) and no effect for self-harm, cognitive functioning, mood/emotion, psychosis as well as hypersentivity/hyperactivity [18]. A second meta-analysis, including the three types of solitary confinement, found numerous deleterious impacts on the mental health of incarcerated people. Inmates in solitary confinement showed a significant increase in mood symptoms (i.e., anxiety and depression) (SMD = 0.41, CI = 0.19; 0.64), psychotic symptoms (SMD = 0.35, CI = 0.18; 0.52), and aggressivity or hostility symptoms (SMD = 0.38, CI = 0.29; 0.47) than individuals not exposed to solitary confinement [19]. Considering that inmates with a mental disorder are more prone to disorganized behavior that can be mistaken for being resistant to regulations, as well as greater difficulty adapting to the correctional facilities environment, studies have evaluated whether they are more predisposed to being placed into solitary confinement than other inmates [20]. In 2020, a meta-analysis including 11 articles with 163,414 inmates showed that inmates with a mental disorder had a 1.62 times higher probability (OR = 1.62, CI = 1.21; 2.15) of being placed in solitary confinement than other inmates without a mental disorder [21].

Unfortunately, these meta-analyses included only administrative confinement or all 3 types of solitary confinement (i.e., administrative, disciplinary, and protective). A deeper understanding of solitary confinement is needed since the types are used for different purposes, the duration and level of restrictions differ, and the process for each type of confinement is different [12,22–24]. Moreover, DC has been the most widely used compared with other forms of confinement in detention [25]. In this sense, some studies have focused on DC but show divergent results. For instance, some studies have observed a higher prevalence of inmates with mental disorders in DC [12,23,26,27]. In contrast, other authors have observed no significant difference [13] or results depending on the type of measure used (e.g., diagnosis, length of observation) [28,29]. Also, the type of measures used to define the presence of a mental disorder (e.g., having been hospitalized in psychiatry in the last year, diagnosis) varies from study to study, making it difficult to interpret the results [27,29]. Concerning the effects on mental health, DC may have a variable effect on different types of mental health symptoms (e.g., anxiety, psychosis, depression) [22,30,31]. It is also possible that the effect may differ for different inmates, particularly whether or not the inmate has a history of a mental disorder [31]. This article is the first systematic review and meta-analysis specifically on DC, covering these topics, and aims to provide a twofold contribution.The first objective was to conduct a systematic review and meta-analysis of the association between mental disorders in inmates and placement into DC in correctional settings compared to inmates without any mental disorder. The second objective was to carry out a systematic review of the effects of DC in correctional settings on the mental health of inmates with or without pre-existing psychiatric conditions.

## Materials and methods

### Search strategies

S.G. and L.D. independently performed a systematic search in the electronic databases of PubMed, PsycINFO, Web of Science, and Google Scholar with keywords that were inclusive for DC (e.g., disciplinary solitary, punitive segregation), psychological effect (e.g., hallucinations, depression, suicide) and mental illness (e.g., psychiatric disorder, mental health). A complete electronic search strategy is available in S1 Table in S1 Text. Reference lists of included manuscripts were screened to ensure at best that no pertinent studies were missed. Searches were completed by October 2023.. No setting, date, or geographical restrictions were applied; searches were limited to English or French language sources.

### Study eligibility

Studies were included if the sample comprised adult inmates over the age of 18 in correctional settings. More specifically, for the first objective (i.e., association between mental illness and placement into DC), the included articles had to report 1) any indication of a potential mental disorder (i.e., mental disorder diagnosed by mental health professionals, taking regular psychotropic medication, received mental health services from a qualified health professional, residency in an

institutional mental health unit) before being place into DC and 2) the statistical associations were calculated/or could be calculated using odds ratios (OR). For the second objective (i.e., effects of DC on mental health), studies were included if a measure of psychological symptomatology or self-harm was incorporated during or after being placed in solitary confinement. Group discussions (S.G, L.D, and A.D) resolved disagreements on including studies to obtain a final consensus (see S1 Table in S1 Text for the electronic search strategy for the systematic review and meta-analysis conducted).

### Data extraction

Data were extracted with a standardized form and double-checked for consistency by the authors. Reported effect sizes with 95% confidence intervals (CI) were recorded with other key information on sample size (e.g., proportion of psychiatric disorder sample, proportion of men), type of psychological symptomatology or indication of a mental disorder, adjustment for confounding factors, type of measure (e.g., administrative data, survey, clinical assessment) and control group. Quality assessment was independently undertaken by S.G. and L.D. against a set of criteria based on the GRADE Checklist for observational studies [32]. Studies were assigned to categories of High, Moderate, Low, and Very Low quality. The details of the retrieved studies are described in supplementary material (see S2 and S3 Tables in S1 Text). Extracted data were independently cross-checked, and any queries were resolved by discussion with A.D. To achieve a high standard of reporting, we followed the Preferred Reporting Items for Systematic Reviews and Meta-Analyses (PRISMA) guidelines (see S4 Table in S1 Text for PRISMA Checklist) [33].

### Statistical analysis

For the meta-analysis (objective 1), data were entered into an electronic database and analyzed in R (version 4.3.2) with the metafor package [34,35]. General population inmates with and without mental disorders were compared to assess whether the risk of being placed in DC was higher for inmates with a mental disorder. The following qualitative descriptions of the strength of reported ORs were used [36]: small (OR = 1.0–1.5), moderate (OR = 1.6–2.5), strong (OR = 2.6–9.9), and very strong (OR = 10.0 and above). Heterogeneity among study point estimates was assessed with the Q statistics and quantified with the $I^2$ index with a value of 0% indicating no effect heterogeneity, whereas 25%, 50%, and 75% suggest low, moderate, and high heterogeneities, respectively [37,38]. Since substantial heterogeneity was observed (see below), random-effects models, which are more conservative than fixed-effects models and appear to address heterogeneity between studies and study samples, were employed [39]. The risk of publication bias was assessed using Egger's test. A significative p-value can indicate publication bias since studies that do not show statistically significant effects may more likely remain unpublished [40]. To have more precision and explain the heterogeneity between articles, sub-analyses were conducted for men and women separately and by four categories of mental health status according to the Diagnostic and Statistical Manual of Mental Disorders, Fifth Edition, Text Revision (DSM-5-TR): 1) severe mental illness (e.g., psychotic disorder, mood disorder), 2) common mental disorder (e.g., anxiety, obsessive-compulsive disorder), 3) mental health services (e.g., having been hospitalized in psychiatry, prescription drug use) and 4) personality disorder [41]. Due to incompatible data, a meta-analysis was not feasible for objective 2; therefore, only the systematic review results are presented.

## Results

### Description of studies

This literature search identified 8,067 potential articles screened for eligibility after removing duplicates. With the search in Google Scholar and cross-referencing, 92 additional studies were identified. Among these articles, 321 full texts were assessed, and 3810were excluded (see file titled *potential articles* in supplementary material for studies identified in the literature and reason for each exclusion). After the assessment, for the first objective (i.e., association between mental

illness and DC), five final articles were included, amounting to 27,455 inmates, and the quality of studies was graded mostly as moderate [13,27–29,42]. One study was undertaken in correctional facilities in England, and the other four in the United States of America. Concerning the second objective (i.e., effects on mental health), six other final articles comprising 171,300 inmates were included [22,26,30,31,43]. Data ranged from low to high-quality evidence. Also, four studies were carried out in correctional facilities in the United States of America, and one was conducted in Denmark. The PRISMA flowchart for including studies in the meta-analysis is found in Fig 1.

## Association of mental health in disciplinary confinement

Incarcerated individuals with a mental disorder were 1.23 times (OR=1.23, CI=1.10; 1.38) more likely to be placed in DC compared to incarcerated individuals without a mental disorder (Fig 2). The database was characterized by moderate heterogeneity (Q=30.38, p=0.016, I²=41.1%). Egger's test (t=0.1429, p=0.78) suggested no publication bias. In both men (OR=1.34, p=0.009) and women (OR=1.24, p=0.002), a significant association between the presence of a mental disorder and placement into DC was observed. By category of mental disorder, there was an increase in the risk of ending up in DC in favor of incarcerated people with a severe mental disorder (OR=1.31, p<0.001) as well as for those with a

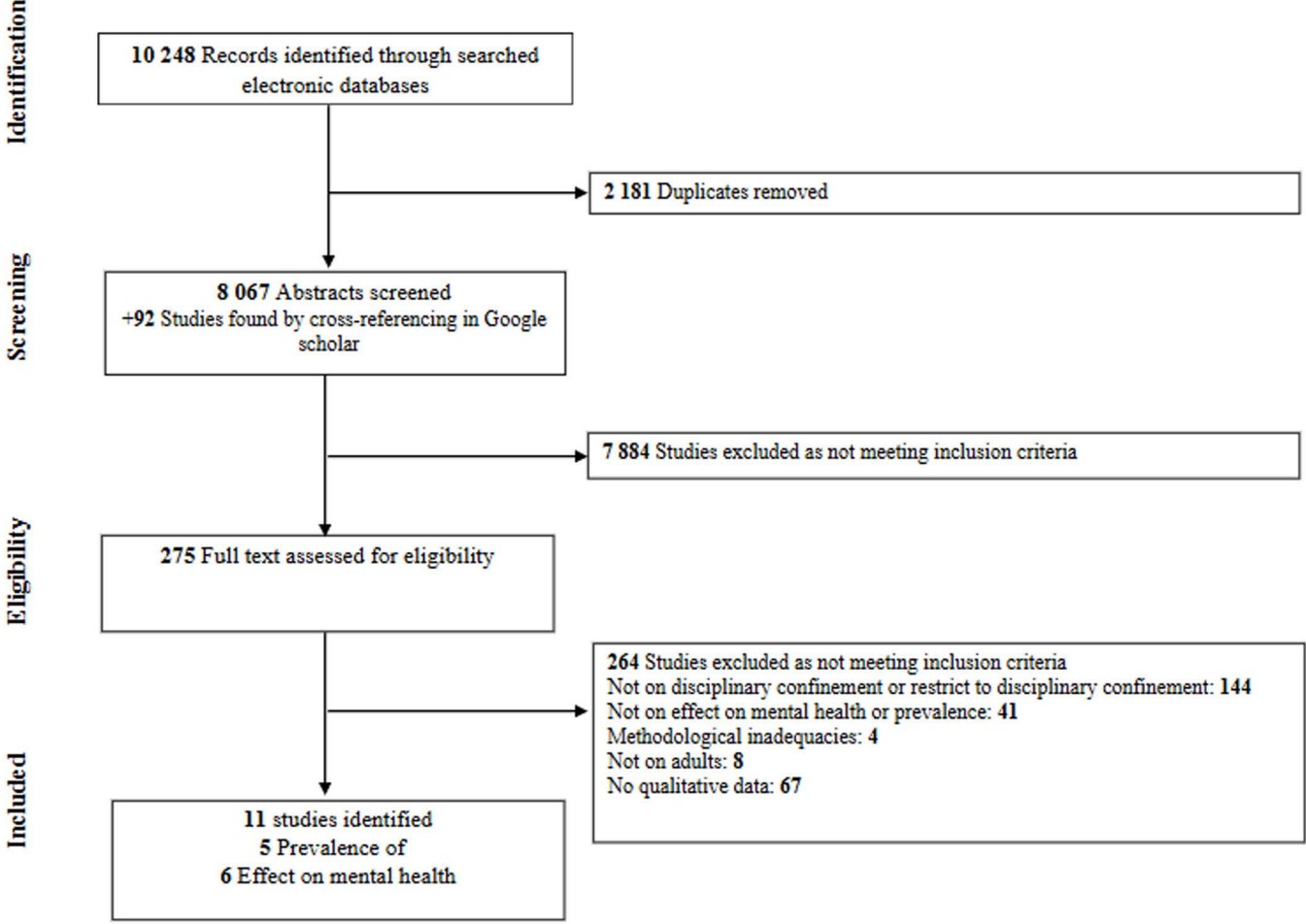

**Fig 1. Flow-chart depicting the search strategy employed to find the studies to include in this review.**

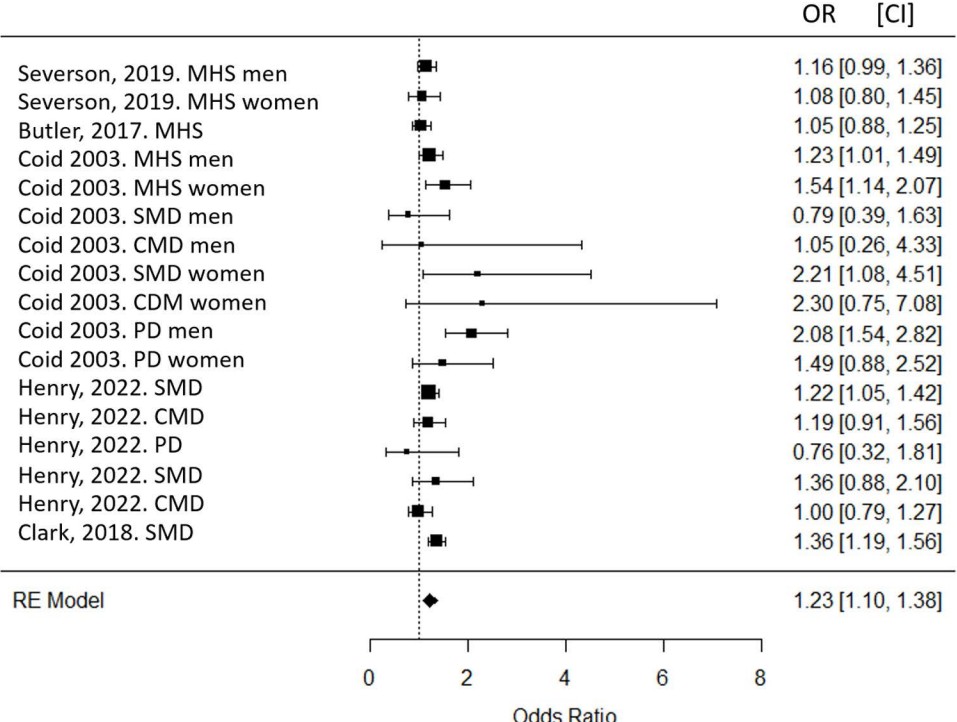

**Fig 2. Forest plot of the association between mental illness and placement into disciplinary confinement in inmates.** SMD: severe mental disorder, CMD: common mental disorder, MHS, mental health services, PD: personality disorder, OR: odds ratio, CI: confidence intervals.

personality disorder (OR=1.66, p<0.001) compared to incarcerated people with no mental disorder. Also, incarcerated individuals who used mental health services before incarceration had an increased risk of ending up in DC (OR=1.16, p=0.024) than those who never used health services. There was no significant association for common mental disorders. Of the five articles included, three controlled for offense leading to DC [27,28,42], and two articles controlled for age, gender, and ethnicity [28,42]. The quality of evidence ranged from low to moderate. Evidence was particularly low for studies that did not control for confounding factors and those that used self-reported measures for diagnosis purposes. The details of the retrieved studies are described in S2 and S3 Tables in S1 Text.

## Effects of disciplinary confinement on mental health

**Psychological distress and psychiatric symptoms.** Two cross-sectional studies conducted by Miller in the United States of America federal penitentiary used clinical interviews to assess psychological distress in 30 inmates who had been in DC (n = 10), administrative confinement (n = 10), or the general correctional population (n = 10). The first study showed that the average distress score assessed with the Brief Symptom Inventory in inmates who were in DC was significantly higher than in administrative segregation and the general correctional population (p < 0.01) [43]. The second study by Miller and Young (1997) observed more obsessive-compulsive symptoms (p = 0.01), interpersonal sensitivity (p = 0.02), and hostility (p = 0.001) in DC compared to inmates in the general correctional population. No significant difference was found for depressive symptoms, anxiety, or paranoid ideation [22].

Another cross-sectional study in the United States of America showed that mentally ill inmates (n = 90) experienced more psychological distress score assessed by survey questionnaire in DC (d = 1.16, p < 0.001) than inmates without mental disorders (n = 85). More specifically, inmates with mental disorders experienced more suicidal ideation, concentration

problems, thought disorders, and perceptual (visual or auditory) disorders in DC compared to inmates without mental disorders. In this study, a mental disorder was a classification given by mental health staff at admission or during incarceration without further specification [30].

The quality of evidence was graded as low for these analyses due to the small sample and the lack of control for confounding factors.

## Mental health services and hospitalizations

A large-scale longitudinal study (n = 36,360) in correctional facilities (i.e., closed prisons, open prisons, and jails) in Denmark found that inmates exposed to DC (n = 4,120) were significantly (p < 0.05) more likely to receive mental health care within three years of incarceration, compared with inmates who had received another type of disciplinary sanction (e.g., fines, warnings, confiscation of contraband) (n = 12,483) and that inmates with no recorded disciplinary actions (n = 19,757) [26]. Mental health care was defined as whether people had contact/consultancy with the general mental health care system after the release and was reported from the administrative database. These analyses were graded as being of moderate quality evidence.

A study with evidence graded as being of low quality has reported from interviews with 55 inmates with no mental disorder before being placed in DC that 40% were transferred to a psychiatric unit due to a deterioration in their mental state, suicide threat or suicide attempt during their placement in DC. In this analysis, there was no control group [30].

## Self- harm

The retrospective study by Kaba et al. (2014) examined predictors of self-harm (defined as self-injurious behaviors not leading to impairment or death) and potentially fatal self-harm (defined as a high likelihood of leading to death or severe impairment) among 134,188 inmates in New York's local detention facilities (New York Jail System). For general self-harm, DC was the most substantial risk factor (OR = 10.15, CI = 8.53; 12.08). The best predictor of potentially fatal self-harm was the combination of exposure to DC and having a severe mental disorder, with an almost 10-fold increase in probability (OR = 9.80, CI = 5.02; 19.18). Severe mental disorders were defined based on the criteria for a psychiatric diagnosis according to the most current Diagnostic and Statistical Manual of Mental Disorders other than alcohol or drug disorders, organic brain syndromes, developmental disabilities, or social conditions and impairment in functioning due to mental illness or reliance on psychiatric treatment, rehabilitation, and supports. The data was collected from the database of the Department of Correction [31]. The quality of evidence of these analyses was graded as high.

The cross-sectional study was conducted in Louisiana's state-operated prisons by Cloud et al. (2023), with evidence evaluated as being of moderate quality compared to self-harm behaviors collected by a survey between inmates in DC (n = 347) and inmates in combined administrative or protective confinement (n = 170). The survey showed that self-harm was 1.97 times more likely (OR=1.97, p = 0.01) to occur in DC than in administrative or protective. This study reported that each 90-day stay in a DC slightly but significantly (OR=1.03, p = 0.02) increased the risk of subsequent self-harm [44].

## Discussion

This study showed a higher risk of inmates with a mental disorder being placed in DC than other inmates, as well as a differential risk depending on the type of mental disorder. Moreover, being exposed to these settings has been shown to cause many adverse effects on the mental state of inmates.

Concerning the first objective, based on the five studies included within our meta-analysis, a significantly higher risk of being placed into DC for any mental health problem within a considerable sample of 27,455 inmates in both men and women was found. Notably, of the five articles included, three controlled for offense leading to DC [27,28,42]. The quality of evidence was graded as being mainly moderate quality. The database was characterized by moderate heterogeneity,

and no publication bias was found. This result aligns with the meta-analysis that included the three types of solitary confinement, which showed a positive association [21]. Also, the results of a study show a significantly higher risk for inmates with a mental disorder of being placed in both short-term and long-term solitary confinement compared to inmates with no mental health issues [45]. Furthermore, inmates with mental disorders are more likely to be placed in solitary confinement short- and long-term twice and more during their detention. Since DC is a process comprising several stages (i.e., 1) receiving a ticket from a guard, 2) penalty decided by a disciplinary committee, and 3) sanction duration), one study evaluated whether the risk for an inmate with a mental disorder differed at each of the three stages. The authors showed a significantly higher risk for inmates with mental disorders at each of the three stages after controlling for the severity of the offense committed [12]. Therefore, inmates with mental disorders received more tickets from guards, they were granted more decisions to be placed within DC, and the duration of confinement was moreover longer than their counterparts without mental disorders. The most significant disparity between mentally disordered and non-mentally disordered inmates was at the first stage when guards handed out tickets [12]. This study proposes that this disproportion of cumulative disadvantage is associated with the stigmatization of individuals with mental illness, which supports the importance of training correctional staff in the management of this population [12,46–48]. A second study, which also controlled for offense severity, likewise showed a higher risk of mentally disordered inmates receiving the most severe form of punishment, DC, than another (i.e., reprimand, lose privilege, higher custody) by the disciplinary committee [42]. Notably, it was shown that a significant risk is linked with severe mental disorders, personality disorders, and having a history of mental health service use. A common mental disorder does not appear to be a risk for placement in DC. Thus, a more pronounced level of disorganization may have increased the risk of being placed in DC. These results are not surprising considering the hypotheses put forward that psychiatric symptoms, such as disorganization, could be perceived as a refusal to obey and would lead to greater difficulty in adapting to correctional facilities [6,7].

Moreover, once placed in DC, inmates may show a degradation of their mental health. The five included studies comprising a large sample of 171,300 inmates showed more significant psychological distress and a greater likelihood of needing psychiatric services than inmates in the general correctional population. A higher risk of self-harm was also reported, and this risk was even higher if the inmates had a pre-existing mental disorder. In addition, two studies that compared solitary confinement to other types of confinement appear to show that DC is more deleterious in terms of self-harm and psychological distress. Although it was not possible to carry out a meta-analysis of DC alone, the results point in the same direction as the meta-analysis that included all three types of solitary confinement, i.e., a positive association between confinement and a deleterious effect on inmates' mental state [19]. Indeed, the authors obtained a small association for mood (d = 0.41, CI = 0.19; 0.64), psychotic (d = 0.35, CI = 0.18; 0.52), and aggressive (d = 0.38, CI = 0.29; 0.47) symptoms. An effect on psychological distress, self-harm as well as the need for psychiatric services and hospitalization have also been observed. Thus, the fact that the duration is predetermined and that a disciplinary process frames it does not seem to mitigate the effect of the DC compared to other types of solitary confinement. Beyond psychiatric symptoms, the DC seems to be associated with a higher risk of suicide. A large study (n = 98,894) in all types of correctional facilities (i.e., prisons for pretrial incarceration and short sentences, standard security prisons for long sentences, high-security prisons for long sentences, juvenile prisons and semi-open prisons) in metropolitan France assessed the number of suicides (i.e., any death resulting from a suicidal act) from administrative database [49]. Results showed, when controlled for age, gender, nationality, education, status, criminal category, main offense, and stage of incarceration, that the suicide risk was 19.9 times higher (HR = 19.9, CI = 14.7; 26.9) during DC than for the rest of the incarceration. Consequently, some states in the United States of America have begun to restrict the number of days allowed and proposed a reform to limit the use of DC [48,50]. Also, studies have set out to find a solution to reduce its use or develop alternative programs [51–53]. For example, a pilot program involving a multidisciplinary team with the aim of reducing exposure to solitary confinement among inmates with a mental disorder showed a decrease in mean rates of disciplinary infractions for assaults, in addition to allowing a positive impact on the health and wellbeing of these inmates [54].

Whereas our meta-analysis shows that inmates with mental disorders are at an elevated risk of being placed into DC and of suffering the effects of these settings, several limitations must be considered. Firstly, few studies specifically targeting DC have been identified in the literature. Moreover, due to the lack of studies, conducting a meta-analysis of the effects on mental health was not possible. Indeed, the effect of DC on each different measure of mental health has only been assessed by one or two studies. One study has examined the effects on inmates with pre-existing disorders, and no study has assessed the possibility of developing a mental disorder following DC exposure. Considering that the risk of being placed in DC varies based on the type of mental disorder and possibly its severity, future research should evaluate how DC could affect the mental state of the inmates by taking into account these factors. Future studies should address these important gaps. Secondly, most of the studies were carried out in the United States of America, thereby reducing the possibility of generalizing the results since confinement conditions may differ between institutions and jurisdictions. Thirdly, most of the results come from self-reported data, which may have underestimated the results for both objectives, given that inmates may lack self-awareness of their illness or be fearful of stigmatization. Finally, several factors, like the level of social isolation and privileges, availability of programming, duration, and the number of confinements, could moderate the effects of DC on mental health [55]. However, these results should be interpreted with caution, given the low quality of evidence in many studies. More studies are needed, with larger sample sizes, that consider these important confounding factors.

This article is important because it contributes to the research, since, to our knowledge, this is the first meta-analysis to look specifically at the risk of being placed in DC for mentally ill inmates, in addition to differentiating between types of mental disorders. In terms of implications for research, this article could guide future studies. For example, it showed the relevance of developing and evaluating alternatives to DC. Also, since the primary aim of DC is to maintain order within correctional facilities, it would be interesting to assess whether it achieves this objective. In addition, these analyses were essential, as they may have implications for future policies that could take into account the deleterious effects of DC on inmates' mental state, as well as the over-representation of mentally ill inmates in DC.

## Conclusions

In conclusion, our meta-analytical investigation showed that mental disorders are at greater risk of being placed in DC (OR=1.23, CI = 1.10; 1.38). More specifically, the risk of being exposed to DC seems to differ depending on the categories of mental health status of inmates: severe mental disorder (OR=1.31, $p < 0.001$), a personality disorder (OR=1.66, $p < 0.001$) and having previously received mental health services (OR=1.16, $p = 0.024$). The results of the systematic review showed that the use of DC has a negative impact on the mental state of the inmates. Thus, more alternatives and policies are crucial since more individuals with mental illness enter correctional systems and solitary confinement has been widely opted as a placement for these inmates. Indeed, changes are needed in the manner correctional facilities manage and care for their inmates with mental health problems. Intervention should be effective in maintaining order and supporting inmates' mental well-being.

## Supporting information

**S1 Text.  S1 Table.** Electronic search strategy for the systematic review and meta-analysis conducted. A search in Google Scholar and bibliographies enabled the identification of additional k = 92 studies. S2 Table. Details of the retrieved studies included. Association between mental disorders of inmates and placement into disciplinary confinement. S3 Table. Details of the retrieved studies included. Effects of disciplinary confinement on the mental health of inmates with or without pre-existing psychiatric conditions. S4 Table. PRISMA Checklist. *From:* Page MJ, McKenzie JE, Bossuyt PM, Boutron I, Hoffmann TC, Mulrow CD, et al. The PRISMA 2020 statement: an updated guideline for reporting systematic reviews. BMJ 2021;372:n71. https://doi.org/10.1136/bmj.n71.
(DOCX)

## Author contributions

**Conceptualization:** Alexandre Dumais.

**Formal analysis:** Charles-Édouard Gigùere.

**Methodology:** Sabrina Giguère, Laura Dellazizzo.

**Writing – original draft:** Sabrina Giguère.

**Writing – review & editing:** Laura Dellazizzo, Alexandre Dumais.

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
