## [Decision Letter · Decision Letter 0]

Dear Dr. Dumais,

Thank you for submitting your manuscript to PLOS ONE. After careful consideration, we feel that it has merit but does not fully meet PLOS ONE’s publication criteria as it currently stands. Therefore, we invite you to submit a revised version of the manuscript that addresses the points raised during the review process.

We look forward to receiving your revised manuscript.

Kind regards,

Andrea Cioffi

Academic Editor

PLOS ONE

Journal Requirements:

4. As required by our policy on Data Availability, please ensure your manuscript or supplementary information includes the following:

Reviewers' comments:

Reviewer's Responses to Questions

**Comments to the Author**

1. Is the manuscript technically sound, and do the data support the conclusions?

Reviewer #1: No

Reviewer #2: Yes

2. Has the statistical analysis been performed appropriately and rigorously?

Reviewer #1: N/A

Reviewer #2: No

3. Have the authors made all data underlying the findings in their manuscript fully available?

Reviewer #1: Yes

Reviewer #2: Yes

4. Is the manuscript presented in an intelligible fashion and written in standard English?

Reviewer #1: No

Reviewer #2: Yes

Reviewer #1: The author is suggested to do revisions as mentioned below:

1) Include only best keywords that describe your paper for indexing and web searches. The more discriminating your keywords are the greater is the likelihood that your article will be found and cited. For example replace “sys-tematic review”

2) The abstract is informative for the purpose, but it do not indicate the significant level of the focus highlighted. Provide more specific reason on how this study will offer a potential route, apart from other works in this area.

3) Personal biases were found at various sections in the paper. The author is suggested to modify the use of "We" in the article. Reform such sentences.

4) Citations are found in the conclusion. References should be generally avoided in conclusions. If you have already cited the ideas earlier in the paper that you are summarizing in your conclusion, you do not need to cite them again.

5) The conclusion derived is not presented well. Modify the conclusion logically.

6) Reformat the acknowledgment section. If Alexandre Dumais (AD) had any other role like funding mention it also.

7) I suggest the author to add some new references by referring more works in the same field, especially the latest works of the year 2024 and indicate some.

8) The motivation and significance on this paper needs to be further emphasized.

Reviewer #2: The work is good but may be improved by the inclusion of the following suggestions.

The “Statistical analysis” section should be added to the manuscript. In the figures ‎the error bar or the standard deviation should be included.‎

Many grammar errors could be found in the paper, please check and deeply revise them. ‎

Please improve the novelty of statement with focus on the novelty.

Quantitative information should be provided in the abstract.

Conclusion section should be rewritten to make it more quantitative.

All abbreviations were defined when used for the first time in the manuscript.

What about Strengths and limitations in your work? Please add it in discussion.

Add your recommendations for future research.

The last paragraph ‎of Introduction should include the study objectives/procedures in brief. ‎

**Do you want your identity to be public for this peer review?** For information about this choice, including consent withdrawal, please see our Privacy Policy

Reviewer #1: **Yes: ** Dr. Abhilash

Reviewer #2: No

---

## [Decision Letter · Decision Letter 1]

Association between mental illness and disciplinary confinement and its effect on mental health: A systematic review and meta-analysis

PONE-D-24-31618R1

Dear Dr. Dumais,

We’re pleased to inform you that your manuscript has been judged scientifically suitable for publication and will be formally accepted for publication once it meets all outstanding technical requirements.

Kind regards,

Andrea Cioffi

Academic Editor

PLOS ONE

Additional Editor Comments (optional):

Reviewers' comments:

Reviewer's Responses to Questions

**Comments to the Author**

Reviewer #2: All comments have been addressed

Reviewer #3: All comments have been addressed

2. Is the manuscript technically sound, and do the data support the conclusions?

Reviewer #2: Yes

Reviewer #3: Yes

3. Has the statistical analysis been performed appropriately and rigorously?

Reviewer #2: Yes

Reviewer #3: Yes

4. Have the authors made all data underlying the findings in their manuscript fully available?

Reviewer #2: Yes

Reviewer #3: Yes

5. Is the manuscript presented in an intelligible fashion and written in standard English?

Reviewer #2: Yes

Reviewer #3: Yes

Reviewer #2: Thanks for your correct revision, and the quality of the paper has been improved greatly so that it is adequate to publish in journal.

Reviewer #3: A clinically relevant systematic review and meta-analysis of the association between mental illness and disciplinary confinement and its effect on mental health.

**Do you want your identity to be public for this peer review?** For information about this choice, including consent withdrawal, please see our Privacy Policy

Reviewer #2: No

Reviewer #3: No

---

## [Editor Report · Acceptance letter]

PONE-D-24-31618R1

PLOS ONE

Dear Dr. Dumais,

I'm pleased to inform you that your manuscript has been deemed suitable for publication in PLOS ONE. Congratulations! Your manuscript is now being handed over to our production team.

Kind regards,

on behalf of

Dr. Andrea Cioffi

Academic Editor

PLOS ONE